# An Overview on ERAP Roles in Infectious Diseases

**DOI:** 10.3390/cells9030720

**Published:** 2020-03-14

**Authors:** Irma Saulle, Chiara Vicentini, Mario Clerici, Mara Biasin

**Affiliations:** 1Cattedra di Immunologia, Dipartimento di Scienze Biomediche e Cliniche L. Sacco”, Università degli Studi di Milano, 20157 Milan, Italy; chiara.vicentini@studenti.unimi.it (C.V.); mara.biasin@unimi.it (M.B.); 2Cattedra di Immunologia, Dipartimento di Fisiopatologia Medico-Chirurgica e dei Trapianti Università degli Studi di Milano, 20122 Milan, Italy; mario.clerici@unimi.it; 3IRCCS Fondazione Don Carlo Gnocchi, 20157 Milan, Italy

**Keywords:** ERAPs, polymorphisms, infection diseases, HIV

## Abstract

Endoplasmic reticulum (ER) aminopeptidases ERAP1 and ERAP2 (ERAPs) are crucial enzymes shaping the major histocompatibility complex I (MHC I) immunopeptidome. In the ER, these enzymes cooperate in trimming the N-terminal residues from precursors peptides, so as to generate optimal-length antigens to fit into the MHC class I groove. Alteration or loss of ERAPs function significantly modify the repertoire of antigens presented by MHC I molecules, severely affecting the activation of both NK and CD8^+^ T cells. It is, therefore, conceivable that variations affecting the presentation of pathogen-derived antigens might result in an inadequate immune response and onset of disease. After the first evidence showing that ERAP1-deficient mice are not able to control Toxoplasma gondii infection, a number of studies have demonstrated that ERAPs are control factors for several infectious organisms. In this review we describe how susceptibility, development, and progression of some infectious diseases may be affected by different ERAPs variants, whose mechanism of action could be exploited for the setting of specific therapeutic approaches.

## 1. Introduction

Even years after the advent of antibiotics and vaccines, infectious diseases remain a significant challenge for public health worldwide, as they are responsible for millions of deaths, mainly among older adults and immunosuppressed or chronically ill people. The potential onset of infectious disease depends on the virulence of the microbe and the efficacy of the host immune response, among others, which envisages the coordinated action of all the elements responsible of defenses against pathogens, from physical barriers to multiple components of the innate and adaptive immune systems. In this setting, CD8^+^ T lymphocytes play a major role in defense and recovery from infections [1]. These lymphocytes recognize immunogenic peptides presented on cells’ surface by major histocompatibility complex (MHC) class I molecules and act through different pathways in order to eliminate the pathogen and the infected cells. Because of the key role played by CD8^+^ T cells in protecting the host from microbial invasions, a multitude of studies investigated how alterations in their pathway of activation can affect the efficiency of the immune response, leading to an increased/reduced susceptibility to infections. MHC class I molecules are directly involved in the process of antigen presentation and CD8^+^ T-cell activation, as they are responsible for binding peptides that have undergone precise structural modifications. Based on these premises, this review aims to provide an overview on the role played by endoplasmic reticulum aminopeptidase 1 (ERAP1) and 2 (ERAP2), which are primarily responsible for the generation of the antigenic repertoire presented by MHC I molecules, in the control of susceptibility as well as progression of different infectious diseases [2,3].

### 1.1. Biological Functions of ERAP1/2

The process through which the human cellular immune system recognizes damaged and infected cells is based on the exposition on their surface of peptides obtained from the proteolytic processing of intracellular and endocytosed proteins, including the aberrant and unneeded ones. These potentially immunogenic peptides are then bound by MHC class I molecules expressed on all nucleated cells and platelets and presented to CD8^+^ T lymphocytes that are capable of initiating the immune response [2].

The shaping of these peptides through the antigen processing machinery (APM) begins in the cytoplasm by the proteasome that cuts intracellular proteins into heterogeneous fragments. Peptides are subsequently transported by the transporter associated with antigen processing (TAP) within the endoplasmic reticulum (ER), where they are trimmed at the N-terminus, in order to generate suitable length peptides to be bound by MHC class I molecules and presented on cells’ surface. In humans, the main aminopeptidases responsible for N-terminal peptide trimming are ERAP1 and ERAP2 (collectively ERAPs), two IFNγ- and TNFα-inducible enzymes that are expressed in various human tissues and belong to the M1 family of zinc aminopeptidases [2]. These two proteins act in concert and play a critical role in shaping conformationally stable peptides to ensure their loading onto MHC class I molecules. The trimming of precursors, indeed, stabilizes MHC class I peptides conformation/structure and presentation [4]. ERAPs activities are coordinated, yet not redundant, indeed, they act together handling substrates and trimming peptides, while maintaining marked differences in enzymatic specificity. ERAP1 preferentially cleaves peptides with large hydrophobic C-terminal amino acids and shows a strong tendency to cut 9–16 amino acids peptides into pieces of 8–9 amino acids, namely the optimal length for binding to MHC I molecules [4,5,6,7,8]. Noteworthy, peptides with proline in position 2 or which are shorter than 8 amino acids are less susceptible to ERAP1 activity [6].

The crystal structure of ERAP1 shows a 4-domain protein whose catalytic site is located in domain II. Domain I caps off the active site and provides binding sites for the amino terminus of a substrate peptide; domain III acts like a hinge enabling conformational changes between domains II and IV; domain IV has a concave structure fronting the active site. Conformational changes in the catalytic site lead to the generation of either an open or a closed conformation [9,10]. The first one confers ERAP1 its receptive properties, owing to which it can bind peptides; the second one allows the protein to close around the peptide and switch to its enzymatically active form [10].

Contrariwise, ERAP2 presents a marked preference for the positively charged arginine and lysine residues situated at the N-terminal and shows a greater efficiency toward shorter peptides that ERAP1 processes poorly [4,11]. Notably, in 2005 L. Saveanu et al. demonstrated that ERAP1 and ERAP2 can physically interact forming a heterodimer [7]. Ten years later, using engineered ERAP1–ERAP2 complexes, I. Evnouchidou and colleagues verified that heterodimer formation changes enzymatic features and increases peptide-trimming efficiency, thus allowing the generation of a wider and more immunogenic antigenic repertoire [8]. Indeed, the complementary activity of the two aminopeptidases leads to the generation of a variegated antigenic repertoire, which may be altered in individuals lacking either protein [12]. Overall, it has been demonstrated that the structural changes made by ERAP1 and ERAP2 represent a key step in mature antigenic epitope generation [12].

Besides its canonical function in acquired immunity, ERAP1 plays a key role in innate immune regulation, as well. Shedding of cytokine receptors [13,14], induction of natural killer (NK) cell development and function (13) nitric oxide formation [15] and triggering of phagocytic activity of splenic DCs and macrophages [16] are some of the functions endorsed by ERAP1 in the initial stages of pathogen recognition.

In addition to their intracellular role, ERAPs are also endowed with some non-immunological functions, such as angiogenesis and blood pressure regulation. These are mediated by the ability of ERAP1 to cleave angiotensin II into angiotensin III and IV, and of ERAP2 to cleave angiotensin III to angiotensin IV [4,17,18]. The localization of angiotensins is extracellular, thus ERAP-mediated cleavage processes can happen only following ERAPs release in the extracellular milieu. In fact, ERAP1 release has been exhaustively demonstrated in immune cells and in murine macrophage cell lines both in soluble and extracellular vesicle (EV) form [19]. A more recent work confirmed that ERAP2 is also secreted by human macrophages triggered by inflammatory stimuli [20].

Even though ERAPs share virtually 50% sequence homology and their function can be fulfilled through heterodimer formation, different studies indicated that the evolution of these two genes is different [4]. Indeed, ERAP1 has a homolog in mice which is called ER aminopeptidase associated with antigen processing (ERAAP), while ERAP2 has no homolog in rodents and possibly it is the result of a rather recent duplication of the ERAP1 gene [21].

### 1.2. ERAPs Polymorphisms (SNPs)

The human ERAPs are encoded by two genes located on chromosome 5q15 in opposite orientation. The ERAP1 gene is 47,379bp in length and consists of 20 exons. The ERAP2 gene is 41,438bp in length and consists of 19 exons [22]. These two genes are highly polymorphic, although ERAP2 gene exhibits not as many single nucleotide polymorphisms (SNPs) as ERAP1. Since these aminopeptidases are involved in the antigen processing pathway, a series of genetic studies have linked their genetic variants to modifications in their functioning that could easily lead to MHC-I-associated disorder onset [12,22,23], including infectious diseases (Table 1 and Figure 1).

#### 1.2.1. ERAP1 Genetic Variants

The majority of disease-related ERAP1 SNPs are positioned near the catalytic site (aa residues 346, 349), within the binding groove (aa residues 725 and 730), or near locations that can influence the conformational rearrangements (aa residues 528 and 575) [9]; other SNPs are also present in interdomain areas or in domain IV, a regulatory region responsible for C-terminal residue peptide binding [9]. Various in vitro and in vivo studies investigating these SNPs have shown that they significantly affect peptide trimming enzymatic activity [24,25,26]. The most prominent ERAP1 SNP is rs30187, encoding the K528R amino acid replacement. Such mutation reduces the peptide trimming efficacy by affecting the kinetic process involved in the modification of ERAP1 from the active to the inactive form [23]. Associated to this SNP, another one, rs27044, which encodes the Q730E amino acid substitution, was demonstrated to be correlated with modifications in peptide length preference as well as trimming specificity [23,24,25]. Remarkably, ERAP1 SNPs can have a synergic impact, combining their effects on the trimming activity. In this regard, analysis investigating the cumulative effect of K528R and D575N (rs10050860) ERAP1 SNPs, showed that an ERAP1 containing K528/N575 displays a much higher enzymatic activity compared to an ERAP1 containing R528/D575 [27]. E. Reeves et al. demonstrated the existence of 13 haplotypes which have been grouped into three categories: efficient, hypoactive, or hyperactive. The allocation was based on the degree of alteration in their ability to generate specific antigenic epitopes [24,28]. The best known ERAP1 SNP combination encodes an allotype with five variants, whose frequency in the European Caucasian population is nearly 26.2%, each of which is independently related to a series of diseases: ERAP1 rs2287987 (V349), rs30187 (R528), N575, rs17482078 (Q725), E730 [25,29]. Intriguingly, this allotype encodes an ERAP1 protein which is non-functional. To further confound this picture, ERAP1 has a co-dominant expression and in heterozygous subjects both allotypes can partake in the global definition of ERAP1 trimming activity in both health and disease. Furthermore, ERAP1 polymorphisms have a wide range of functional degrees that can affect either mRNA synthesis as well as ERAP1 enzymatic activity, which in turn can influence disease onset or susceptibility. Notwithstanding, the strong linkage disequilibrium (LD) among numerous polymorphisms presumably leads to a restricted variability among individuals.

In the field of infectious disease, Y. Yao et al. have identified several SNPs which influence disease susceptibility and/or progression. These polymorphisms have been subdivided in intronic region like rs2248374, rs1748133, rs149481, rs27042 rs149173 and exon regions as I276, R127, N392, L848 and are exhaustively described in ref [30].

#### 1.2.2. ERAP2 Genetic Variants

Only few variants have been so far identified in ERAP2 sequence. The rs2549782 SNP, encoding the K392N aminoacidic substitution, occurs in most populations with a balanced frequency and modulates the enzymatic activity and substrate specificity of the aminopeptidase [31]. Indeed, in vitro studies showed that the N392 variant is much more effective than its 392K counterpart in trimming hydrophobic N-terminal residues [30]. Such difference is secondary to changes in the catalytic site and in the binding site of the N-terminal domain. This SNP can lead to a variability in the antigen processing activity among individuals carrying either variant of the enzyme [30]. Rs2549782 is in LD with another polymorphism, the non-coding rs2248374 (A/G). Notably, the G allele of this polymorphism is transcribed into a spliced form with an extended exon 10 (56 extra nucleotides) containing two in frame TAG stop codons, which undergoes degradation through nonsense-mediated decay (NMD) process [32]. Notably, SNP rs2248374 is in LD with 11 other SNP variants; this allows the definition of two haplotypes, hereafter referred to as HapA (A allele for rs2248374) and HapB (G allele for rs2248374), at intermediate frequency in most human populations [31]. The maintenance of both haplotypes with the same incidence suggests that, despite NMD, even transcript(s) derived from HapB could confer fitness advantages, possibly in particular milieus. Indeed, an analysis recently conducted on dendritic cells carrying the G allele of this SNP revealed that two previously unknown short isoforms (ERAP2/ISO3, ERAP2/ISO4) are transcribed from this haplotype following influenza infection, thanks to the creation of a different preferred splice site [33]. This finding suggests how the balancing selection of the rs2248374-tagged haplotypes could be the consequence of variability in ERAP2 expression—including the transcription of alternative forms of the protein—and peptide processing, providing an advantage against viral infections [33].

As a consequence of the strong LD between the G allele of rs2248374 and the N392 variant of the rs2549782 SNP, K392 allotype is almost exclusively present in a number of different populations. An exception is represented by Chileans, where the two SNPs were not found in LD, thereby emphasizing the complexity of the ERAP2 locus in different populations [34]. Furthermore, it is conceivable that individuals who are homozygous for HapB—about 25%—do not produce the protein; conversely, the ones who are homozygous for the HapA haplotype present about 50% more enzyme in comparison to individuals with the heterozygous haplotype. Although the total absence of ERAP2 protein seems not to disable the cellular life and more broadly the immune system response, further studies are needed to identify the molecular reasons behind the maintenance of such apparently useless haplotype at 50% frequency [34].

Another polymorphism with a considerable effect on ERAP2 protein expression is rs10044354 (C/T), where homozygosity for the C allele results in very decreased levels of ERAP2 [34,35]. Recently, a study associated the polymorphism rs75862629 (A/G) in the promoter region of the ERAP2 gene with changes in the expression of ERAP2 and ERAP1 in the Sardinian population [36]. In particular, the G allele of rs7586269 resulted in a decreased ERAP2 protein expression and a higher ERAP1 expression, suggesting the possibility of a concerted regulation of the expression of both genes by this polymorphism.

## 2. ERAPs and Infectious Diseases

Microbes and their mammalian hosts have co-evolved for millions of years, resulting in intricate host–pathogen interactions [37]. Infected cells progressively digest intracellular microbial proteins and display the resulting antigenic repertoire on cell surface molecules for recognition by T cells [37]. As this process initiates cell-mediated immune responses and is essential to detect infections, genetic variants of proteins involved in the antigen presentation pathway play an important role in shaping the susceptibility and response to invading pathogens [38]. In turn, different microbes have evolved sophisticated mechanisms to evade recognition by the immune system [39]. Given the key role played by ERAPs in antigen processing and presentation, it is conceivable that these enzymes may be potential targets and modulators of the pathogenicity of infectious diseases. Viruses, in particular, are obligate intracellular parasites, whose antigenic peptides are expressed on the surface of infected cells in association with MHC I molecules, after being processed inside the host cell (see paragraph 1.1). In this review we will discuss the effect of ERAPs in modulating viral infections including HCV, HCMV, influenza virus, HPV, and HIV (Table 1 and Figure 1). Surprisingly, new data have recently associated the activity of ERAPs also with infections mediated by extracellular pathogens (bacteria and parasites) whose antigenic peptides, presented in association with MHC class II molecules, are generated through the endosomial pathway independently from ERAPs processing. Such findings, briefly discussed in paragraph 2.6, once more underline that ERAPs activity may rely on mechanisms independent from antigen presentation.

### 2.1. ERAPs and Hepatitis C Virus (HCV)

Hepatitis C virus (HCV) infection—for which a preventive vaccine is not yet available and which has infected about 71 million people worldwide—is one of the major causes of acute and chronic liver diseases and results in chronic hepatitis, liver cirrhosis, and hepatocellular carcinoma (HCC). A percentage of people who contract the infection does not develop chronic hepatitis, spontaneously clearing the virus; conversely about 30% of chronically infected individuals develops one of the above-mentioned conditions [40]. It has been demonstrated that the evolution of HCV infection is dependent on age at infection, gender, alcohol abuse, obesity, and insulin resistance, with a great role played by genetic factors [41]. The latter, when related to the immune response, may lead to different susceptibility to HCV pathogenesis among individuals. Fortunately, the availability of highly efficient direct-acting antivirals has revolutionized the treatment landscape with over 95% cure rates as accurately reviewed in [42].

The HCV life cycle is still far from being fully understood because of the difficulties in developing an in vitro model that can be used to analyze thoroughly the viral replication and viral entry mechanism [41,43]. Concisely: while circulating in the bloodstream, the HCV virion, covered in host low-density lipoproteins, interacts with a variety of receptor molecules expressed on target cells, thus eliciting a clathrin-mediated endocytosis process. In the cytoplasm, the disruption of the viral capsid allows the release of the single-stranded positively polarized RNA genome. The genomic RNA is then translated into a polyprotein precursor within the ER and then cut into ten single proteins by viral and host proteases [40,44]. The assembling of a replication complex leads to the synthesis and of new virions, which leave the cell by exocytosis through a Golgi-dependent pathway [40].

An effective immune system aims to eliminate the infected cells owing to the presentation on their surface of antigenic peptides, bound and exposed by MHC class I molecules. As ERAP1 and ERAP2 play a critical role in shaping these suitable-sized peptides, polymorphisms in their gene sequence can significantly affect the produced antigenic repertoire, thus conditioning infection chronicity as demonstrated for HCV infection [4,11,45].

In a recent study, S. Liu et al. investigated the impact of ERAP1 and ERAP2 variants on HCV chronic infection in a cohort of 376 Han Chinese with chronic HCV infection and 324 healthy controls [46]. Allelic and genotypic frequencies of rs27044, rs30187, rs26618, and rs26653 in the ERAP1 gene and rs2248374 and rs2549782 in the ERAP2 gene were compared between the case and control groups. Notably, the allelic distribution of rs26618 highly deviated in HCV-infected patients compared to controls, revealing that the C-allele carriers had an increased risk of susceptibility to HCV chronicity compared to the T-allele carriers. According to these results, the authors assumed that the rs26618 polymorphism may affect either the enzymatic activity or the structure of ERAP1, leading to alterations in the HCV antigen presentation pathway, and in turn to increased susceptibility to this infection. The allelic distribution of rs2248374 was different in the case and control groups as well. In particular, the A allele caused a slightly higher risk of susceptibility to HCV chronicity compared to the G allele. As previously reported, the G-allele for this SNP leads to NMD and hampers ERAP2 protein expression [32]. On the other hand, individuals who are homozygous for the A-allele present about 50% more enzyme in comparison to individuals with the heterozygous haplotype, suggesting that rs2248374 may be a marker of ERAP2 protein expression. Taking this into account, this study suggests that the A-allele for this SNP may be associated with increased expression of ERAP2 in HCV chronically infected individuals, therefore affecting the antigen presentation process and leading to a greater risk of susceptibility to HCV chronicity [36]. In addition, given the role of ERAP2 as a promoter of ERAP1 activity—through the formation of heterodimers with increased trimming efficacy—it has been observed that combination of rs26618-C/rs2248374-A on the two genes also provides a greater risk of susceptibility to HCV chronicity in Han Chinese population [46].

Another recent case-control study assessed the impact of ERAP1 allotypes on virus-specific CD8^+^ T-cell response in a human leukocyte antigen (HLA)-B*27:05^+^ individual with acute HCV-infection [47]. Despite the patient displayed favorable prognostic indicators—symptomatic course of infection, IL28B/IFNL4 CC genotype and HLA-B*27 expression—he did not succeed in clearing the virus and developed persistent low-level infection with HCV [47]. In-depth analyses showed that this individual carried two hypoactive ERAP1 allotypes and his CD8^+^ T cells targeted subdominant epitopes longer than the classical 9-mer ones, never described previously. The authors, therefore, concluded that ERAP1 allotype deeply affect the virus specific CD8^+^ T cell repertoire in vivo, shaping epitopes of different lengths which elicit qualitatively diverse CD8^+^ T-cell responses. Although there was no direct evidence of a cause-and-effect correlation between the hypoactive ERAP1 variants and the generation of an alternative antigenic repertoire, it is known that ERAP1 peptide trimming activity is modulated by ERAP allotypes, which in turn condition HCV-specific CD8^+^ T-cells recruitment. Taken together these results suggest an association between the process of antigen shaping by ERAP1 and the outcome of HCV infection [47].

### 2.2. ERAPs and Influenza Virus (Flu)

Influenza virus is the etiological agent of influenza, an acute respiratory disease yearly affecting millions of people worldwide. Mostly, influenza virus causes seasonal epidemics of disease with peak illness periods which have a heavy impact on worker and school absenteeism and productivity loss. The majority of individuals who contract the infection require a relatively brief time to recover from fever and other typical symptoms; however, in other cases influenza results in more serious illness and possibly death, particularly among high-risk groups, causing about 290,000 to 650,000 deceases per year [48]. Every year a new mostly effective vaccine is available and recommended, in order to prevent the disease in healthy adults and to reduce infection severity among the elderly.

There are four different types of influenza viruses: A, B, C, and D, even though the latter has not been demonstrated to cause illness in humans yet. Influenza virus binds to the epithelial cell receptors and enters through an endocytosis process [49]. Once inside the target cell, the viral RNA is released into the cytoplasm and then transported into the nucleus, where transcription and replication occur. Among the resulting viral proteins, some combine with the replicated RNA fragments generating new virions [49]. Others proteins are instead degraded by the proteasome into peptides that are transported within the ER where they undergo further modifications driven by ERAP1 and ERAP2, before being loaded onto MHC class I molecules [49].

Given the known link between HLA-B27 and ERAPs in conferring genetic predisposition to ankylosing spondylitis (AS) [50,51], A. Akram et al. performed a study on double transgenic (Tg) mice (i.e., B27/ERAP−/− and B7/ERAP−/−), in order to further investigate the correlation between ERAP and different MHC I alleles. In particular, considering that flu immunodominant epitopes (nucleoprotein (NP) 383–391 for HLA-B27^+^ and NP418–426 for HLA-B7^+^) are well characterized, the study was conducted on flu-infected mice. Results demonstrated that ERAP activity is deeply influenced by its MHC-I context, highlighting how significant is the cooperation between HLA-B27 and ERAP in peptide shaping and presentation. In fact, while in association with HLA-B7 the presence or absence of ERAP did not affect the immune response, its lack in combination with HLA-B27 resulted into a substantial decrease of CTL response toward NP383-391 epitope during flu infection. Such defect was possibly secondary to a decreased ability to trim the N-terminally extended NP383–391 peptides in absence of ERAP, resulting in a reduced quantity of B27/NP383-specific CD8^+^ T cells in B27/ERAP−/− mice [49]. On the other hand, the lack of alterations of B7/NP418 CTL responses in flu-infected B7/ERAAP−/− mice suggest that ERAP is dispensable in B7 epitope generation [52].

More recently, considering the role of ERAP2 in peptide shaping and presentation, C.J. Ye et al. investigated the generation of alternative isoforms of this gene in response to influenza infection, examining RNA-seq profiles of monocyte-derived dendritic cells from 243 individuals [33]. First, they showed that the rs2248374-G allele reduces ERAP2 expression both in resting and IFNB1-stimulated cells. Moreover, in flu-infected cells the presence of two previously unknown short isoforms (ERAP2/ISO3, ERAP2/ISO4) was observed. Remarkably, following IFNB1 stimulation the two short isoforms were not any more detectable, suggesting that their transcription is possibly dependent on viral stimuli, upstream of interferon 1 signaling pathway. The two short isoforms, transcribed from HapB, are different from the full-length one – ERAP2/Iso1, transcribed from HapA —because their transcription begins in correspondence of exon 9 and undergoes alternative splicing of an extended exon 10. Moreover, they differ from each other by alternative splicing at a secondary splice site at exon 15 and they are both translated into proteins lacking the catalytic domain [33]. Previous studies had already demonstrated that full-length ERAP2 tends to form hetero- and homodimers with ERAP1, in order to perform its trimming activity in a more efficient way [12]. Therefore, the author speculates that the expression of the two short flu-specific isoforms may exert a dominant negative effect on wild type (WT) ERAP1 or ERAP2. This would lead to an altered peptide processing, possibly conferring an advantage against infection by presenting a more immunogenic antigen repertoire [12] (Figure 2). Both haplotypes—and relative transcripts—may therefore provide fitness advantage in diverse environmental conditions, providing a plausible explanation to HAPB maintenance at intermediate frequencies by balancing selection [12]. This observation evidently opens new prospective in the field of antigen presentation and immune system regulation that will be worthwhile pursuing.

### 2.3. ERAPs and Human Cytomegalovirus (HCMV)

Human cytomegalovirus (HCMV) is a beta-herpesvirus that causes infection in 40% to almost 100% of individuals worldwide, depending on socio-economic and geographical factors [53]. Following infection the virus generally remains silent in bone marrow myeloid cells, causing an asymptomatic infection which can undergo sporadic reactivation [54]. HCMV infection, though, can be life-threatening and fatal in immunocompromised individuals [55,56,57]. Moreover, in intrauterine HCMV infection, the developing fetus can be easily infected leading to a variety of birth defects [55,56,57]. HCMV infections are typically associated to salivary glands, even though the virus can live and replicate in plenty of different cells, thus facilitating its dissemination throughout the body [58]. A vaccine is not available yet, but its development has been defined to be a priority by the Institute of Medicine and the National Vaccine Advisory Board (Washington, USA), because of the prevalence and the relevance of this virus [59].

HCMV life cycle starts with the binding of trimeric or pentameric viral complex to host receptors which differ in fibroblasts (PDGFRα) and in epithelial cells/endothelial cells, (Nrp2 or OR14I1). Such interaction allows viral entry into the cell by fusion or endocytosis [60]. The viral capsid is then transported into the nucleus, where transcription, replication, and encapsidation can take place [61]. At the same time, some viral-encoded proteins intervene in the control of cell-signaling pathways and cellular metabolic processes to facilitate viral replication on one hand, immune evasion on the other [62,63]. The subsequent formation of a nuclear egress complex, together with the rupture of the nuclear lamina, leads to capsid transfer into the cytoplasm, previously assembled within the nucleus [64,65,66,67]. Virions are then shaped and trafficked to the viral assembly complex (AC), which includes components of the endoplasmic reticulum, Golgi apparatus, and endosomal machinery [68,69,70]. Once capsids gain tegument and viral envelope at the AC, the new infectious particles are released together with the non-infectious ones, the so called “dense bodies.”

The existence of a link between ERAPs and HCMV infection is strongly suggested by analyses performed on viral microRNA (miRNA). MiRNAs are short RNA segments—with a length between 19 and 23 nucleotides—capable of regulating gene expression by completely or partially pairing with their target mRNA. This leads to mRNA cleavage, destabilization, or translational inhibition [71]. CMV has been shown to express at least 24 miRNAs targeting both viral and host genes implicated in immune response [72]. S. Kim et al. specifically performed an analysis on HCMV miR-US4-1 and demonstrated that it targets ERAP1 mRNA, leading to its destabilization or degradation, therefore preventing the production of this enzyme [72]. Endorsing this finding, ERAP1 levels were shown to be inversely proportional to miR-US4-1 expression. This, in turn, affects the shaping of HCMV-derived immunogenic peptides and translates into viral evasion from CD8^+^ T cells-mediated immune response [72]. Such result suggests that HCMV developed a way to specifically and preferentially target ERAP1—despite the presence of other aminopeptidases involved in antigen presentation (i.e., ERAP2 and placental leucine aminopeptidase)—likely because of its leading role in peptide generation, its more ubiquitous presence, and overall higher expression [72].

A more recent study focused onto another HCMV miRNA, miR-UL112-5p, whose function resembles the one played by miR-US4-1. Indeed, miR-UL112-5p directly targets the 3’ UTR of ERAP1-A SNP variant, thus downregulating its mRNA expression and altering the presentation of many HCMV-derived antigenic peptides to CD8^+^ T cells [73]. Notably, the authors identified the rs17481334-G SNP variant, naturally occurring in the ERAP1 3’ UTR, which disrupts the consensus sequence for HCMV miR-UL112-5p and, in turn, prevents ERAP1 targeting and degradation [73]. Consistently, ERAP1 expression was significantly decreased by miR-UL112-5p overexpression only in fibroblasts from AA individuals, while fibroblasts from GG individuals did not show any alteration in RNA and protein levels [73]. Even more important, a significantly reduced HCMV seropositivity was detected among GG individuals suffering from multiple sclerosis, a disease model in which HCMV infection is negatively associated with adult-onset disorder [73].

Besides describing an important HCMV immune evasion strategy, these results provide some evidence that person to person variation in susceptibility to infection could be governed by polymorphisms in gene sequence recognized by miRNA, at least for certain pathogens.

### 2.4. ERAPs and Human Papilloma Virus (HPV)

Human papilloma virus (HPV) is a DNA virus and the causative agent of cervical cancer. HPV is the most common sexually transmitted infection and indeed almost the totality of population undergoes this infection sooner or later during life [74]. HPV infection is asymptomatic and spontaneously resolves within a couple of years in the majority of cases; however, in a minority of cases the infection worsens, resulting in warts or precancerous lesions, possibly resulting in cervical as well as in anus, vulva, vagina, penis, and oropharynx cancers [74]. Actually, HPV is responsible for more than 12% of cancers in women in underdeveloped countries [74]. Three vaccines are currently available as prevention methods: they all are thoroughly effective in preventing HPV infection, especially toward types 16 and 18 viruses, responsible for more than half of cervical cancer cases worldwide [75].

HPV targets epithelial cells of the basal layer in injured regions. HPV infection and growth are absolutely dependent upon the expression of the complete program of keratinocyte differentiation as thoroughly described in Human Papilloma Virus’ Life Cycle and Carcinogenesis [76]. In fact, there is a strong dependence of viral lifecycle progression on host cells differentiation pathway, considering that new virions egress from terminally differentiated cells of the most superficial layer of the epithelium. As these cells can no longer replicate, HPV synthetizes the E6 and E7oncogenes—two viral proteins engaged on cell cycle control and genome maintenance—which, in turn, inactivate p53 and retinoblastoma protein (pRb), responsible for controlling healthy cell lifecycle [77].

Conversely, the expression of the viral genes essential for viral replication is highly accelerated in the more superficial layers of the epithelium, this results in viral genome amplification and in the production of thousands of viral copies per cell [78,79,80].

In order to identify the molecular pathway on which to intervene to develop effective tumor vaccines and/or immunotherapies, a variety of studies focused on the role of APM and HLA in shaping and presenting HPV-derived peptides, thus eliciting T cells immune response and the consequent lysis of cervical carcinoma cells [81].

In 2009, A.M. Mehta et al. genotyped 12 nonsynonymous, coding low molecular weight peptide (LMP)2, LMP7, TAP1, TAP2, and ERAP1 polymorphisms in 127 cervical carcinoma patients and 124 controls [82]. They identified a specific combination of four SNPs in genes encoding components of the APM (ERAP1-127, ERAP1-730, TAP2-651, and LMP7-145), which were associated with an increased cervical carcinoma risk [82]. In addition, they found that up to 12% of all cervical carcinomas correlated with a particular haplotype combination that include the minor allele at the ERAP1-127 and ERAP1-730 loci, and the major allele at the TAP2-651 and LMP7-145 loci. This haplotype results in a three-fold increase in cervical carcinoma risk. Implications and functional roles of these polymorphisms were not investigated in this study, although the authors hypothesize that they could be responsible for a different modulation of gene expression or for a change in protein functionality or, alternatively, could reflect a causative linkage with other, still unknown gene loci [82].

The same authors, in a subsequent study investigated APM and HLA class I expression in 109 cervical carcinoma patients and for the first time reported a downregulation in ERAP1 expression in an HPV-derived cancerous lesion. Indeed, the protein was shown to be a strong predictor of decreased overall survival, probably because its downregulation may cause a preferential presentation of non-tumor-associated antigens, thus promoting tumor progression [83].

In 2012 a study conducted by A. Hasim et al. investigated the role of ten genes within the HLA-I and APM family on HPV-dependent cervical cancerous lesions in a cohort of Uighur women [84]. The aim was to analyze the correlation between cancer development and genetic variants or aberrant regulation of HLA-I and APM, including gene promoter methylation, transcription and protein expression. Results showed that cancerous lesion development correlates with the partial or total loss of HLA-I, TAP1, TAP2, LMP2, LMP7, ERAP1, tapasin, calreticulin, and ERp57 mRNA and protein expression [84].

In order to deepen the role of APM in cervical carcinoma, another study on the occurrence of twelve nonsynonymous coding polymorphisms in the LMP2, LMP7, TAP1, TAP2, and ERAP1 genes in relation to disease outcome has been conducted by A.M. Mehta et al. on 75 cervical carcinoma cases [85]. Notably, this study first proved a correlation between ERAP1 polymorphisms and its protein expression. Actually, results showed the ERAP1-127 SNP and the ERAP1-56–ERAP1-127 haplotype to be highly correlated to ERAP1 expression variability. Moreover, homozygosity and heterozygosity for ERAP1-127 SNP, as well as the ERAP1-56–ERAP1-127 haplotype were associated with decreased and increased overall survival, respectively. As the ERAP1-127 SNP maps in the peptidase M1 domain of the protein, the authors suggested that such SNP could be responsible for modifications in ERAP1 proteolytic function, leading to the presentation of an altered antigen repertoire, which would facilitate immune escape. Alternatively, the genetic variations at the analyzed loci may not have a direct effect on ERAP1 activity, but they could have a remote effect on other loci in LD [85].

To further investigate the contribution of genetic variations in the APM to HPV-induced cervical carcinoma risk, A.M Mehta et al. in 2015 conducted a further study on the genotype and genotype interactions of 11 previously identified SNPs in seven genes coding for some APM components, including ERAP1 [86]. The analysis was based on the occurrence of these variants in Javanese and Balinese Indonesian populations, in whom HPV is almost endemic [87]. Results showed that, as in the Dutch population, APM genetic variability is correlated with cervical carcinoma risk, but patterns of association differ between the two populations. In particular, while in Javanese a strong association between cervical carcinoma risk and ERAP1-575 locus on chromosome 5 and the TAP2-379 and TAP2-651 SNPs on chromosome 6 could be established, only the TAP2-651 locus correlated with carcinoma risk in Balineses [87]. The authors hypothesized that this diversity may be due to differences in the genetic composition of these two populations, which reflect two distinct historical-geographic contexts [88,89]. Indeed, these two Indonesian areas show a different oncogenic HPV subtypes distribution suggesting that distinct HPV subtypes may evade the immune system by hijacking different host genetic factors, providing also an explanation for the distinct association patterns in the Dutch population [90].

In a subsequent study, A.M. Mehta et al. investigated the causes of ERAP1 downregulation in cervical carcinoma. Thus, they analyzed ERAP1 mRNA expression in tumor cells where the level of ERAP1 protein was known to be downregulated, comparing them to cells from patients with normal ERAP1 levels. Notably, ERAP1 mRNA level was demonstrated to be significantly correlated to decreased protein production. This finding suggests that the causes leading to protein downregulation occur at a pre-transcriptional level and may include mutations as well as complete and partial loss of heterozygosity and haplotype loss. These data represent the first understanding of the in vivo processes responsible for ERAP1 protein downregulation in cervical carcinoma [85].

Recently, S. Steinbach et al. performed a comprehensive analysis concerning the expression of the APM in a panel of HPV16-positive cell lines, focusing on both mRNA and protein levels [91]. Results showed an upregulation of different mRNA molecules in the APM pathway. Conversely, at protein level only ERAP1 synthesis was significantly increased in HPV16-positive cell lines and in cervical intraepithelial neoplasia and cervical cancer lesions. These results did not mirror the ones reported by A.M. Mehta and colleagues [83]; however, they complied with the Human Protein Atlas, which indicates cervical cancer as a condition where ERAP1 is upregulated [92]. Since ERAP1 overexpression had already been associated with the elimination of immunodominant epitopes from melanoma [93] and colorectal carcinoma [94,95], the study focused on defining the role of ERAP1 upregulation in antigen presentation in HPV16-induced carcinomas. The authors speculated that a progressive increase of ERAP1 level in disease progression is paralleled by a decreased presentation of E7 81–91 and other epitopes, leading to a lack of targets for the immune system, perfectly functioning by itself [94,95].

The most recent study seeking possible correlations between HPV infection and ERAP1 has been conducted in 2019 by E. Reeves et al. whose work aimed at investigating the association between ERAP1 allotypes and the degree of lymphocyte infiltration in HPV-induced tumors [96]. In general, lymphocyte infiltrations—specifically by CD8^+^ T cells—are related to better clinical outcome in various cancers [97,98,99,100,101,102]. In some HPV-derived cancers—cervical squamous cell carcinoma (CSCC) and oropharyngeal squamous cell carcinoma (OPSCC)—the specificity of tumor-infiltrating CD8^+^ T lymphocytes (CD8^+^TILs) for HPV epitopes has been demonstrated. In particular, analyses showed that peripheral and CD8^+^ TILs recognize epitopes originating from the E6/E7 proteins expressed by the viral genome [101,103,104]. Moreover, a correlation between TIL infiltrates and a better survival was shown in OPSCC [101,105]. Because of ERAP1’s pivotal role in the antigen processing pathway, the authors verified if ERAP1 allotypes correlate with the amount of CD8^+^ TILs in 22 patients with HPV^+^ OPSCC, grouped into three different categories (low, intermediate, high CD8^+^ TIL levels). Although many allotypes were shared between the CD8^+^ TIL groups, some differences were observed in their trimming capacity. Indeed, ERAP1 allotype pairs in the CD8^+^ TIL^high^ group had a major trimming capacity compared to the one expressed by individuals with low CD8^+^ TIL numbers. This finding suggests the idea that ERAP1 function correlates with the quality of anti-HPV T-cell response which, in turn, modulates tumor outcome [98].

To our knowledge, the only data associating ERAP2 SNPs to HPV-induced cervical cancer are those by L. Chuanyin et al. In particular, they established a correlation between rs26653 and rs27044 in ERAP1 and rs2287988 in ERAP2 with cervical intraepithelial neoplasia (CIN) and cervical cancer, suggesting that different ERAP SNPs may have combinatorial effects on disease susceptibility. However, it is worthwhile to be mentioned that screening for HPV-infection was not performed in the analyzed cohort, therefore the correlation between these ERAPs SNPs and HPV-infection is merely deductive 10.21203/rs.2.21433/v1 BMC Cancer (under review).

A more complete and exhaustive understanding of the role of APM SNPs and haplotypes in HPV-mediated carcinogenesis is likely to be a key step in the development of novel tumor vaccines and immunotherapies.

### 2.5. ERAPs and Human Immunodeficiency Virus (HIV)

HIV belongs to the lentivirus subfamily of retrovirus and is the causative agent of acquired immunodeficiency syndrome (AIDS). An estimated 1.7 million individuals worldwide became newly infected with HIV in 2018; 38 million people worldwide are living with HIV and the only treatment option for these patients is antiretroviral therapy [106]. The presence of the virus causes a severe immunodeficiency condition, characterized by CD4^+^ T cell loss responsible for the onset of infections and the development of malignant tumors that otherwise rarely infect human beings to cause illness [107]. Two different types of HIV have been identified: HIV type 1 and HIV type 2 which differ in genome structure and geographical distribution but both cause AIDS syndrome [107].

Briefly, HIV life cycle consists of four stages. HIV viral entry is characterized by the gp120 binding to the cell surface via the CD4^+^ receptor and the CCR5/CXCR4 co-receptors. Once inside, the viral reverse transcriptase converts RNA genome into a single stranded (ss) DNA copy which is immediately replicated to produce a double stranded (ds) molecule. The viral genome is then transported into the nucleus, integrated into the host cell genome by viral integrase. The proviruses may remain transcriptionally inactive for a long time and this event is partially responsible for the latency of HIV in infected people. Finally, proviral polyproteins are translated and cleaved by the protease and new infective virions are generated [107].

Since ERAPs shape the final repertoire of peptide antigens presented by MHC class I molecules to CD8^+^ T cells, the two aminopeptidases have been investigated as potential targets and/or modulators of HIV-1 infection and virus-host interaction. One of the first studies in this field showed the involvement of ERAPs in the creation of a specific HIV-1 CTL response hierarchy and epitope abundance [108]. Actually, structural and functional analyses demonstrated that antigen cleavage process modulates the number and length of epitope-containing peptides, thereby affecting the response avidity and clonality of T cells. Another link between ERAPs and HIV-infection was evidenced in 2004 by R. Draenert et al. [109]. The authors demonstrated that in HLA-B57^+^ HIV- infected individuals there is a mutation at residue 146 (alanine to proline) of the HIV Gag protein immediately before the NH2 terminus of a dominant HLA-B57-restricted CTL epitope. This mutation was found to prevent the NH2-terminal cleavage by ERAP1, leading to decreased HIV-specific CD8^+^ T cell immune response. Notably, these results show that allele-associated sequence alteration within the flanking region of CTL epitopes can modify antigen processing and are consistent with the observation that HLA-B57^+^ patients often exhibit a strong and protective CTL response [109].

In 2010, R. Cagliani et al. demonstrated an association between ERAPs polymorphisms and HIV-1 infection [110]. Thus, the authors showed that long-standing balancing selection has maintained genetic variability at human ERAP1 and ERAP2 genes. Additional results indicated that the rs2549782-G ERAP2 polymorphism is significantly more represented in an Italian cohort of HESN (HIV exposed seronegative individuals) subjects who, despite repeated exposure to HIV-1 infection, do not seroconvert [110]. M. Biasin et al. further confirmed such association in another HESN Spanish cohort comprising intravenous drug users [111]. Moreover, HLA-B typing indicated that the HLA-B*57 allele, which has been associated with both delayed progression to AIDS and decreased susceptibility to HIV-infection, is significantly more common than expected among HESN homozygous for rs2549782-G [111]. Notably, by analyzing HIV-infected peripheral blood mononuclear cells (PBMCs) from subjects carrying different ERAP2 alleles the authors also detected an alteration of the antigen processing/presenting machinery, which presumably results into a quantitative and/or qualitative variation in MHC class I complexes presented on target cells [111].

In another study, J. Dinter et al. compared the capacity of monocyte-derived macrophages (MDMs) dendritic cells (DCs) and monocytes to produce HIV-derived epitope precursors and epitopes [112]. Results showed that the expression of cytosolic proteases involved in antigen processing, including ERAP1 and ERAP2, is higher in MDMs than in DCs. As a consequence, the kinetics and amount of antigenic peptide produced as well as the intracellular stability of HIV peptides prior to MHC loading significantly increased in MDMs, leading the authors to conclude that this can result into variations in the timing and effectiveness of recognition of HIV infected cells by CTLs [112].

L. Chen et al. demonstrated that ERAP1 silencing reduced the presentation of the HIV-Gag immunodominant HLA-B27 epitope, KK10 [113]. Furthermore, they observed that cells expressing the Arg528 ERAP1 variant, which has an in vitro slower substrate-trimming rate compared with the WT one, were less susceptible to CTL lysis compared to those expressing WT ERAP1, despite equivalent amount of ERAP1 expression. Overall, these results suggest that ERAP1 silencing or the presence of less efficient ERAP variants reduce functional presentation of a naturally occurring HLA-B27 epitope to CD8^+^ T cells, therefore modifying the antigen repertoire and activation of the immune system [113].

In addition, interesting results showed that ERAP2 polymorphisms (rs2549782 and rs2248374) strongly associate with hypersensitivity syndrome to nevirapine in subjects carrying the HLA-C*04:01 allele. Notably, this report is the first one to investigate the interaction between drug-induced HLA disease and the ERAP gene [114].

Finally, we have recently shown that ERAP2 as well as ERAP1 may be secreted from MDM under inflammatory stimuli [19,20]. Notably, the addition of recombinant human (rh)ERAP2 to PBMC drastically reduced HIV infection leading to an increased T-cell activation and a decreased percentage of terminal differentiated CD8^+^ T lymphocyte (TEMRA) [20]. The mechanism by which exogenous ERAP2 interferes with HIV-1 infection/replication is under investigation. However, the observation that HIV replication is partially reduced by rhERAP2 addition even when cell cultures are CD8^+^ T cells-depleted suggests that such an effect is not exclusively dependent on CD8^+^ T cells [20].

### 2.6. ERAPs and other Microrganisms

Other infections have been at least partially associated to ERAP gene expression and/or function and are worthwhile to be mentioned.

Lymphocytic choriomeningitis virus (LCMV) infection intensely triggers the immune system and is almost entirely controlled by CD8^+^ CTLs, which proliferate massively upon LCMV infection [115]. In 2006, I. A. York et al. using mice infected by LCMV, described a role for ERAP1 in shaping and handling peptides within the ER. In particular, they reported that the hierarchy of LCMV-derived immunodominant epitopes was extremely different in WT and ERAP1-KO infected mice [116]. While some LCMV epitopes were immunodominant in ERAP1-KO mice, they were not present in WT mice, because of the trimming activity of ERAP1 provoking their degradation [116]. On the other hand, LCMV-derived epitopes normally processed by ERAP1 were not present in ERAP1-KO mice. These results highlight the importance of ERAP1 protein in the immune response to the virus and its role in altering antigen shaping as well as epitope abundance on the cell surface [116]. Of note, one year later, Firat and co-workers reported that despite the marked reduction of MHC class I expression levels (40%) in ERAP1 KO mice, they did not detect significant differences, upon acute LCMV infection, in the percentage of splenic CD8^+^ T cells in ERAP1 KO mice compared with WT ones. The authors therefore concluded that owing to the plasticity of the T-cell response, only severe variances in epitope presentation may translate into significantly different CTL frequencies [117]. Exploiting the same ERAP1-KO mice model, J. Yan et al. observed that ERAP1 expression can exert a positive, neutral, or negative effect in the generation of distinct class I epitopes. Indeed, when ERAP1-KO mice were infected LMCV, a drastic reduction of antigen epitope presentation (more than 80%) was observed. Conversely, against several influenza-derived epitopes they observed only a slightly reduced CTL response, but this did not reach statistical significance [118]. Overall, these partly inconsistent findings suggest that ERAPs could exert a different contribution in the generation of antigens hierarchy in different viral infections. Thus, the consequences of ERAPs deficiency could be particularly evident in cases in which the response relies only on very few antigens whose presentation is significantly diminished by ERAP1 or low frequencies of specific CTLs are induced.

In 2011, F-J Tsai et al. conducted a study to identify which genetic factors could predispose to Kawasaki disease (KD), whose etiology has been related to bacteria, viruses or other still poorly defined environmental factors [119]. They performed a genome-wide association study (GWAS) on 250 KD patients and 446 controls in a Han Chinese population settled in Taiwan and later replicated the analysis on 208 cases and 366 controls from an independent Han Chinese cohort. Notably, two polymorphisms, rs149481 and rs27042, located in an ERAP1 gene intron were found to be associated with KD. These results underline how ERAP1 trimming function may have an important role in activating the immune response in KD, possibly modifying the peptide repertoire of a still unidentified pathogen [119].

In 2008, N. Blanchard et al. conducted a study on antigen presentation in a murine model of *Toxoplasma gondii* infection [120]. Results showed that the immune response to *T. gondii* infection is mediated by CD8^+^ T cells specific for an immunodominant and protective decapeptide, HF10, presented by the H-2Ld MHC I, whose generation is critically dependent on ERAAP proteolysis. Actually, ERAAP-deficient mice were extremely susceptible to infection and died rapidly, thus assigning to ERAAP the role of “susceptibility factor” for toxoplasmosis in mice. Certainly, this analysis of H-2d mice opens doors to further studies, aiming to provide the basis for the development of a *T. gondii* vaccine for humans [120].

Two years later (2010), T.G. Tan et al. in an attempt to make the previously mentioned study applicable to a human context, focused on the orthologous gene of ERAAP in humans, i.e., ERAP1 [121]. The authors aimed to verify whether any polymorphisms of the protein could be correlated to congenital toxoplasmosis. The analysis was performed on a North American cohort which included 124 congenitally infected children: on a total of 32 genotyped SNPs, two of them—rs149173 and rs17481856, in strong LD with each other—were found to be associated with susceptibility to human congenital toxoplasmosis; no correlation with ERAP2 SNPs was found in this study [121].

E. Lorente and co-workers performed a study on N-terminally extended precursors of naturally processed HLA-B27 antigens from human respiratory syncytial virus (HRSV) demonstrating that ERAPs work in concert, each trimming the product generated by the other as substrate for further N-terminal cleavage [122].

Studies on in vitro cell-based antigen presentation showed that the high ankylosing spondylitis-risk ERAP1 allotype (Met349/Lys528/Asp575/Arg725/Gln730) destroyed more quickly most of HLA-B27 peptides, whether from viral, bacterial or self-origin [123]. Therefore, the authors assumed that the co-expression of HLA-B27 molecules and ERAP1 allotypes with greater enzymatic activity modifies the ordinary presentation of pathogen and self-peptides to the acquired immune system setting the conditions to autoimmunity [123].

Even more recently, analyses of the microbioma of ERAP1−/− mice showed a significant enrichment for Cyanobacteria and Actinobacteria, mainly Prevotella, Odoribacter, Bacteroides, as compared to the microbioma of similarly housed and age-matched WT mice [124]. These findings led to establish a correlation between a role for ERAP1 role in immunodominance and gut dysbiosis in ERAP1−/− mice, possibly because of altered immune tolerance and resulting in the intestinal colonization by aberrant microbial communities. Further studies investigating the possible role of ERAPs variants in intestinal dysbiosis are needed.

## 3. Conclusions

In this review, we have summarized the recent knowledge on the biology of ERAP1 and ERAP2 enzymes and their possible links to several infectious diseases. Polymorphic variations in *ERAP1* and *ERAP2* genes, modifies their function and their skill to produce antigenic peptides and controls cytotoxic responses against antigen-presenting cells. The numerous studies reviewed in this manuscript definitely confirm that such variations in ERAPs expression and/or genetic variants have an important repercussion on the onset and/or progression of infectious diseases. Indeed, ERAPs function as key factors within the antigen presentation pathway, influencing the interaction between pathogens and natural host resistance.

Expression of HLA-class I risk alleles and their relation with ERAPs involves unusual ER peptide processing leading to altered peptide presentation as a mechanism responsible for increased susceptibility or progression of several infectious diseases. Loss of ERAPs function has also been demonstrated to significantly affect the presentation of epitopes by MHC class I molecules, possibly contributing to the maintenance of chronic infections. However, how ERAPs are functionally associated with infectious diseases, and how ERAPs interact with disease-associated MHC class I molecules, are still unanswered questions. Additional functional studies employing proper comparative analysis and functional confirmation assays in different populations are, therefore, necessary to better understand how ERAP variants can affect infectious disease predisposition and pathogenesis. The investigation of epistasis interaction between ERAPs and HLA variants is also essential in order to properly disclose the mechanism through which ERAP1 and ERAP2 influences infectious diseases and the process of antigen presentation.

A number of infectious conditions are characterized by an alteration of both the innate as well as the adaptive immune system, which would be sensitive to ERAP manipulation. While the role played by ERAPs in acquired immunity has been extensively investigated, only few studies have analyzed ERAP1 participation in natural immunity [13,14,16,19], and data on the possible contribution of ERAP2 to innate immunity are almost missing. Further studies in this direction could therefore provide novel and unexpected insights to clarify the molecular mechanism by which different ERAPs variants modulate the onset, reactivation, and development of infectious diseases. Likewise, the recent identification of two new ERAP2 isoforms following influenza virus infection [33] justifies the maintenance of a haplotype (HapB) whose value was so far unknown. Similar analyses will be needed to clarify the possible involvement of this haplotype in the control of other microbial infections. Furthermore, considering the concerted action of the two enzymes leading to enhanced trimming activity, even polymorphisms that may affect the quantity and quality of the heterodimers formed should be taken into consideration. Likewise, the two new ERAP2 isoforms produced as a consequence of influenza virus infection could modify dimer interaction exerting a negative effect on the wild type forms [33]. Similar analyses will be needed to clarify the possible involvement of this haplotype in the control of other microbial infections.

It is absolutely mandatory to unveil the role and function of key components of the class I antigen presentation complex to understand how the human immune system provides a primary and, in most cases, efficient line of defense against infecting agents. The findings so far associating ERAPs functioning to the world of infectious diseases and more recently to natural occurring intestinal dysbioses support the possibility of targeting ERAPs through pharmacological or genetic modifications in order to provide novel immunotherapies for controlling microbial communities and infections.

## Figures and Tables

**Figure 1 cells-09-00720-f001:**
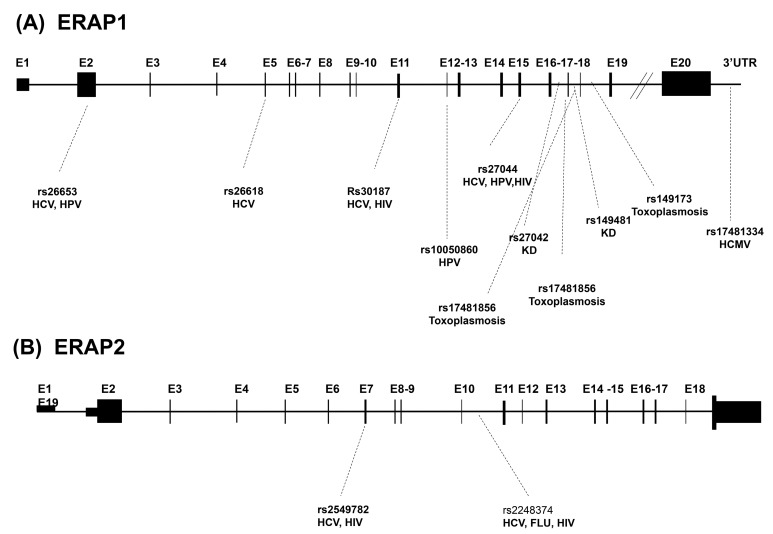
Genomic structure of the human ERAP1 (**A**) and ERAP2 (**B**) genes. Exons are numbered and depicted as boxes. Outlines indicate the sites of exonic or intronic polymorphisms described in the text as rs and related infections.

**Figure 2 cells-09-00720-f002:**
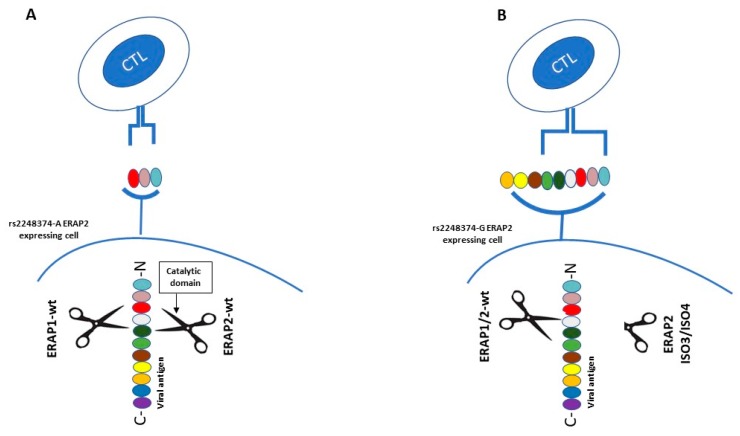
Schematic representation of the mechanism of action displayed by ERAP2 ISO3 and ISO4 in the antigen presentation pathway. (**A**). Following virus infection rs2248374-A ERAP2 expressing cells produce wild type ERAP2 (ERAP2-wt) which can homodimerize or heterodimerize with ERAP1-wt (ERAP2-wt^+^ERAP2-wt; ERAP1-wt^+^ERAP2-wt), in order to process viral antigens to be presented on cell surface for recognition by specific cytotoxic T lymphocyte (CTL) clones. (**B**). Following virus infection rs2248374-G ERAP2 expressing cells may transcribe two alternative spliced isoforms: ERAP2-ISO3 and ERAP2-ISO4. This two variants, unlike ERAP2-wt, lack the catalytic domain but can still heterodimerize with both ERAP2-wt and ERAP1-wt. As a result, these unconventional heterodimers (ISO3^+^ERAP2-wt; ISO3^+^ERAP1-wt; ISO4^+^ERAP2-wt; ISO4^+^ERAP1-wt) may process viral antigens differently from the canonical ones, generating an alternative antigenic repertoire. This in turn may activate other CTL clones and more broadly a more or less protective response by the immune system.

**Table 1 cells-09-00720-t001:** A list of endoplasmic reticulum aminopeptidase 1 (ERAP1) (**A**) and ERAP2 (**B**) sequence variants involved in infectious diseases.

**(A)**			
**ERAP1 SNPs**	**Region**	**Variation**	**Infectious disease(s)**
**rs30187**	Exon 11	K528R	HCV, HIV
**rs27044**	Exon 15	Q730E	HCV, HIV, HPV
**rs10050860**	Exon 12	D575N	HPV
**rs26618**	Exon 5	M276I	HCV
**rs26653**	Exon 2	P127R	HCV, HPV
**rs17481856**	Exon 17	L848L	Toxoplasmosis
**rs17481334**	3’ UTR	None	HCMV
**rs149481**	Intron 17	None	KD
**rs27042**	Intron 16	None	KD
**rs149173**	Intron 18	None	Toxoplasmosis
**rs17481856**	Intron 17	None	Toxoplasmosis
**(B)**			
**ERAP2** **SNPs**	**Region**	**Variation**	**Infectious disease(s)**
**rs2549782**	Exon 7	K392N	HCV, HIV
**rs2248374**	Intron 10		HCV, Influenza, HIV

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
