# Peer review of "An Overview on ERAP Roles in Infectious Diseases"

_cells, 2020, doi:10.3390/cells9030720_

Round 1
Reviewer 1 Report
The review entitled "An overview on ERAP roles in infectious diseases" describes how ERAP variants could influence the development and progression of different infectious diseases. Although the manuscript focuses primarily on viral infections, the evidences are well organized and presented. The literature cited within the manuscript is adequately complete and up-to-date. The review may be of interest, since only limited works specifically address the role of ERAP immunopeptidase in infectious diseases. Overall the manuscript was well written with only minor editorial and formatting corrections. While I support the publication of this manuscript in Cells, I strongly recommend that the following comments be addressed prior to publication.
- The Figure 2 is not sufficiently explanatory. The authors should make more evident the difference between the conditions described in panel A and panel B.
- In the Conclusions section, the author should discuss more extensively the value of the presented studies, in order to highlight the current impact and possible future developments of the reported discoveries.
- Line 372-377. Authors should better clarify the meaning of AA individuals and GG individuals.
- Line 424-426. The authors should reword the sentence as unclear.
- The bibliography should be reviewed, as it contains many Italian entries and is not consistent with the style of the bibliography.
Reviewer 2 Report
The manuscript by Saulle et al. describes the roles played by endoplasmic reticulum aminopeptidase 1 and 2 (ERAP1 and 2) in the anti-pathogen immune response, specifically in the cases of infection by CMV, influenza virus, HPV and HIV. There have been few reviews on ERAPs recently but none that covered this topic as well and as specifically as this one. The review is well-organized and studies are cleverly discussed. The analysis of the litterature is thorough and provides a panoramic landscape of the field.
Here are some minor considerations:
Too many references are reviews instead of original findings. This should be addressed. References format is not uniform and dates are not in English.
One format of citing X. XXXX et al., should be chosen and standardised throughout the text.
Table 1 could be 'prettier' if formatted differently
Figure 1 is low resolution and should be refered to earlier in the text. A subfigure containing the SNPs described in the text that are not related to infectious diseases could be added. A 3D structure image or a simple representation of the proteins could help with the visualisation of the reported mechanisms.
Figure 2 could be improved. Are the peptides longer in case B when ISO3 and 4 are expressed? Is there still some catalytical activity? why is half of a scissor drawn?
The + in CD8+ T cells should be "superscripted".
Here are some other comments:
Line 29: Other factors than virulence and host cell immune response may impact on the onset of infectious disease. The sentence 'The potential onset of infectious disease depends on the virulence of the microbe and the efficacy of the host immune response which envisages the coordinated action of all the elements responsible of defenses against pathogens, from physical barriers to multiple components of the innate and adaptive immune systems' could be modified not to exclude other possibilities.
Line 42: 'above mentioned changes' refer to the abstract and could be repeated in the main text.
Line 46: add 'cellular' between 'human' and 'immune'
Line 56: [In humans, ... aminopeptidases] should mention the original papers.
Line 61: [The trimming ... presentation ] add reference to original discovery.
Reference 1.2.3.4..... are reviews
Line 68. Phrasing is confusing in :[The crystal structure of ERAP1 shows a 4-domain protein whose catalytic site is located in domain II a cavity, covering areas of domains II and IV, is the substrate-binding site.]
Line 90: Ambiguous phrasing: confirm that it may. [A more recent work confirmed that also ERAP2 may be secreted by human macrophages triggered by inflammatory stimuli]
Line 138: Phrasing.
Line 150: could you be more specific? [Such observation suggests that, despite NMD, transcripts derived from both haplotypes could confer fitness advantages, possibly in diverse milieus]
Reference 38 and 39 should cite original findings.
Section 2.1 ERAPs and HCV. The existence of efficient anti-HCV drugs is worth mentioning.
Line 245: The role of ERAP2 on ERAP1 function should be further (and earlier) discussed? [In addition, given the role of ERAP2 as a promoter of ERAP1 activity - through the formation of heterodimers with increased trimming efficacy - it has been observed that combination of rs26618-C/rs2248374-A on the two genes also provides a greater risk of susceptibility to HCV chronicity in Han Chinese population (46).]
Line 254 missing space
Line 264: the onset of the flu?
Line 270: The sentence [An effective vaccine is available...among the elderly.] may be misleading: even though vaccination is recommended (and should be) and is indeed beneficialm the sentence does not take into account that each year the flu vaccines are different, as is their efficacy.
Line 292: standardise FLU or flu.
Line 274: The sentence should refer to original findings.
Line 337-338 Which country of the [Institute of Medicine and the National Vaccine Advisory Board]
Line 242: uniform the police.
Line 346-347 ref 60 and 61 are reviews.
Line 374: Can you define the nomenclature of AA and GG individuals.
Line 391: missing space
Line 393: Its not [It's...]
Line 405: APM and HLA abbreviations could have been defined ealier, line 52 or 240?
Line 408: Define LMP
Line 448: [highly] could be removed
Line 492: [shered] is shared
Line 513: phrasing could be better.
Line 519: latency can also be explained by other independant mechanisms, the sentence : [The proviruses may remain transcriptionally-inactive for a long time and this event explains the latency of HIV ininfected people.] should be less affirmative.
Line 530: are these findings consistent with the fact that HLA-B57+ patients often exhibit a strong and protective CTL response?
Line 551: Nevirapine
Line 570: rhERAP2 = rhesus?
Line 609: underline.
What are the mechanisms if the SNPs are in introns?
This last part of the review (2.6 ERAPs and other microorganisms) could be better structured.
Overall, great review.
